# Sex Disparity for Patients with Cutaneous Squamous Cell Carcinoma of the Head and Neck: A Systematic Review

**DOI:** 10.3390/cancers14235830

**Published:** 2022-11-26

**Authors:** Brandon Tan, Ishith Seth, Olivia Fischer, Lyndel Hewitt, Geoffrey Melville, Gabriella Bulloch, Bruce Ashford

**Affiliations:** 1Research Central, Illawarra Shoalhaven Local Health District, Wollongong Hospital, Wollongong, NSW 2500, Australia; 2Central Clinical School, Monash University, Melbourne, VIC 3004, Australia; 3Illawarra Health and Medical Research Institute, Wollongong, NSW 2522, Australia; 4Faculty of Science, Medicine and Health, University of Wollongong, Wollongong, NSW 2522, Australia; 5Sydney Head and Neck Cancer Institute, Chris O’Brien Lifehouse, 119-143 Missenden Road, Camperdown, NSW 2050, Australia

**Keywords:** head & neck, squamous cell carcinoma, surgery, cutaneous

## Abstract

**Simple Summary:**

This is a systematic and meta-analysis of proportions study that investigated the distribution of head and neck cutaneous squamous cell carcinoma between men and women. Males were disproportionally affected by HNcSCC overall, and this was most evident in the ear subtype, while this disparity was less obvious for the eyelid subtype. This paper underscores the risk of metastatic progression of HNcSCC and denotes further research is warranted to elucidate the mechanisms for these sex differences. Improving our understanding of this will better inform clinical practices, and eventually improve patient outcomes for all—males and females.

**Abstract:**

The incidence of head and neck cutaneous squamous cell carcinoma (HNcSCC) is unevenly distributed between men and women. At present, the mechanism behind this disparity remains elusive. This study conducted a systematic review and meta-analysis of proportions to investigate the disparity between sexes for patients with HNcSCC. PubMed, Scopus, EMBASE, MEDLINE, Emcare and CINAHL were searched in November 2021 and June 2022 (N > 50, English, human), and studies which examined the association between sex and HNcSCC were included. Analysis was conducted using RStudio with data and forest plots displaying males as a proportion of total patients with HNcSCC. Two independent researchers performed study selection, data extraction, data analysis and risk of bias. Eighty-two studies (1948 to 2018) comprising approximately 186,000 participants (67% male, 33% female) from 29 countries were included. Significantly more males had HNcSCC overall (71%; CI: 67–74). Males were also significantly more affected by cSCC of the ear (92%; CI: 89–94), lip (74%; CI: 66–81), and eyelid (56%; CI: 51–62). This study found HNcSCC disproportionately affected males overall and across all subtypes. Improving our understanding of sex-specific mechanisms in HNcSCC will better inform our preventive, therapeutic and prognostic practices.

## 1. Introduction

Cutaneous squamous cell carcinoma (cSCC) is the second most common cancer besides basal cell carcinoma (BCC) and is the most common cancer with metastatic potential [1]. BCC and cSCC are collectively known as non-melanoma skin cancer (NMSC), which are the most commonly diagnosed cancers worldwide [2] and carry the highest financial burden of all cancers in Australia [3]. NMSC causes significant burden to health services worldwide and negatively impacts the patient’s quality of life, given that most tumours are on sites with high sunlight exposure and visibility, such as the head and neck, which require excision and impose cosmetic concern [4]. Ultraviolet (UV) radiation-induced somatic mutations are considered an early event for cSCC carcinogenesis, and immunosuppression is a key risk factor for their development [5]. This risk factor lends to being the most common cancer among immunocompromised individuals [6]. Metastasises are found in up to 5% of cases, and their five-year survival is 70% [7]. Despite this high incidence, its prevalence is said to be underreported [2,8]. Furthermore, as cSCC appears to have a predilection for older individuals [9], the burden on the healthcare system will likely increase as the population ages.

cSCC is known to disproportionately affect men, although the mechanisms behind this disparity remain elusive [10]. The anatomical distribution of cSCC between men and women, where men are more prone to head and neck cSCC (HNcSCC) and women lower limb cSCC, lends itself to several hypotheses for this phenomenon [9]. For example, men are less compliant with sun protection measures [11] and more likely to perform outdoor work [12], although a laboratory mice study shows male rats had more aggressive and clinically poorer outcomes histologically when cSCC tumour grade and overall risk profiles were matched between sexes [6]. Emerging evidence also demonstrates that males are disproportionately affected for cancer incidence, have poorer treatment responses, and have worse outcomes even when adjusting for known epidemiological risk factors [13,14,15].

Little is known about whether non-environmental factors can partly explain the male predominance for HNcSCC However, the growing body of literature has proven some sexual dimorphism in mutational responses and immune system activation during cancer carcinogenesis [15]. Whilst theories about the protective effects that female have regarding patterns of genetic alteration, genomic expression [13,14] and possible stronger immunoediting have been hypothesised [15], the evidence supporting these is lacking.

Therefore, this systematic review and meta-analysis of proportions was designed to investigate sex disparities in patients with a diagnosis of HNcSCC. This paper aimed to identify gaps in our understanding of this disparity to be the basis of future research and actions by public health.

## 2. Materials and Methods

This systematic review was prospectively registered with PROSPERO, the international prospective register of systematic reviews (http://www.crd.york.ac.uk/prospero (accessed on 23 November 2022), registration no. CRD42022298850, registered on the 16.01.2022). This study was conducted according to the Preferred Reporting Items for Systematic Reviews and Meta-Analyses (PRISMA) statement (https://www.prisma-statement.org/ (accessed on 23 November 2022)).

### 2.1. Search Strategy

The electronic databases of PubMed, Scopus, EMBASE, MEDLINE, Emcare and CINAHL were searched in November 2021 and then again in June 2022. The search was limited to the English language and research conducted on human subjects. The search string for Scopus was: (TITLE-ABS-KEY ((head OR neck) AND (cancer OR neoplasm)) AND TITLE-ABS-KEY (sex OR gender) AND TITLE-ABS-KEY (“squamous cell skin carcinoma”) AND TITLE-ABS-KEY (cutaneous OR skin)) AND (LIMIT-TO (LANGUAGE, ”English”)) AND (LIMIT-TO (SRCTYPE, “j”)). Duplicates were removed, and two independent researchers (BT, OF) searched the output to determine the studies that were eligible for inclusion. The references of the full-text articles were also reviewed for relevant studies. Differences in the selection of articles were discussed with the wider research team to reach a decision. Data were extracted independently by two researchers and compared for consistency. Differences in the extracted data were also discussed and resolved.

### 2.2. Inclusion and Exclusion Criteria

An article was included if it: (1) included humans; (2) contained quantitative research and had been published in an English-language, peer-reviewed journal; (3) related to participants with a head and neck cutaneous squamous cell carcinoma; (4) examined the association between sex and cutaneous squamous cell carcinoma of the head and neck; (5) was an observational or experimental study with a minimum sample size of 50 participants. An article was excluded if: (1) it was a study investigating cSCC outside of the head and neck region; (2) there was no analysis regarding the association between sex and cSCC; (3) Other types of cancer, even if they occurred in the head and neck region; (4) Grey literature (e.g., book chapters, conference abstracts, dissertations).

### 2.3. Population, Intervention, Comparator, and Outcome

The participants studied in this review were those diagnosed with head and neck cutaneous SCC. The exposure was head and neck cutaneous squamous cell carcinoma, and the comparator was sex (male, female). Sex was defined anatomically and from what was reported in the demographic and analysis sections of the paper being reviewed. There were no restrictions placed on the type of setting the participants were from, nor on the comorbidities of the patient population. The primary outcome was the association between sex and cutaneous squamous cell carcinoma occurring in the head and neck (HNcSCC). The secondary outcome was to determine factors (e.g., differences in genomic sequences between males and females) that were reported to contribute to or support this association.

### 2.4. Data Extraction

Extracted data included the first author’s name, publication year, country, study design, time-period reviewed by the study, sample size, characteristics of participants, setting, type of cancer, the incidence of metastasis, and the association between sex and HNcSCC. Subgroup analyses (eyelid, ear, lips) were completed for only those studies that had more than 50 participants with that condition.

### 2.5. Statistical Analyses

Statistical analysis of proportions was conducted using RStudio (RStudio 2022.07.1 + 554 [2022], https://www.rstudio.com/about (accessed on 23 November 2022)). The gender proportion of cSCC data was estimated using the R function metaprop with inverse-variance weighting. I2 values of 50% or more indicated substantial heterogeneity. Thus, the random-effects model was used in all analyses. Proportions were logit transformed. Data and forest plots display males as a proportion of total patients with cSCC with a 95% confidence interval. A *p* value < 0.050 was considered statistically significant.

### 2.6. Quality Assessment

Quality assessment was completed using the NIH Quality Assessment Tool for Observational Cohort and Cross-Sectional Studies (https://www.nhlbi.nih.gov/health-topics/study-quality-assessment-tools (accessed on 23 November 2022)). Internal validity questions were answered by two independent reviewers as yes, no, not reported, cannot determine or not applicable. A quality rating (Appendix A) was then categorised as poor (0–4), fair (5–10) or good (11–14) [16].

## 3. Results

### 3.1. Description of Studies

A primary search of the databases retrieved 679 studies. After duplicates were removed, 546 studies were available for the title and abstract screening, after which 182 studies were eligible for full-text screening. Following this, 62 studies were available for analysis from the database search. In addition, citation searches identified 356 studies, and 72 were assessed for eligibility. Twenty of these were added to the study. Finally, 82 studies were available for systematic review and meta-analysis. The reasons for exclusions are shown in Figure 1 and the 82 studies are shown in Figure 2, Figure 3, Figure 4 and Figure 5.

Across these 82 studies, a total of 186,000 participants (67% male, 33% female) from 29 countries. Studies included were observational and were from the United States (22.5%), Australia (17.5%) and Europe (17.5%). Most studies generally (80%) investigated cSCC of the head and neck, and the remaining studies concentrated HNcSCCs at specific anatomical sites (e.g., ear, eyelid, or lip). The included studies reported data from the years 1948 to 2018. A summary of the included studies is reported in Appendix A and the risk of bias is reported in Appendix A. No studies investigated genomic sequences as direct outcome measures.

### 3.2. The Proportion of Males with cSCC of the Head and Neck

The association of sex with overall HNcSCC was reported in 70 studies with a male predominance of 70.57% (95% CI = 67.04–73.86%, *p* < 0.001). Studies investigating HNcSCC of the lip (*n* = 13), eyelid (*n* = 13) and ear (*n* = 13) found males were also disproportionately affected (for cSCC of lip males = 74.09%, 95% CI = 66.39–80.54%, *p* < 0.001; for cSCC of ear males = 92.01%, 95% CI = 89.05–94.22%, *p* < 0.001). This was also true for cSCC of the eyelid; however, the model estimated a lower proportion than other subgroup analyses (males = 56.33%, 95% CI = 50.54–61.95 %, *p* = 0.032). These results are demonstrated in Figure 2, Figure 3, Figure 4 and Figure 5.

### 3.3. Overall Incidence

Studies report that the overall incidence (per 100,000 persons) of cSCC occurring in the head and neck is higher in males [77,78,79,80,81], with relative tumour density on the scalp being ten times higher than in females and five times higher on the ears [82]. Poorly differentiated cSCCs were most frequently found on the scalp, cheek/chin, forehead and ear for men and the cheek/chin for women [83]. HNcSCC lesions also accounted for a larger proportion of all invasive lesions in men [84]. The relative risk of being diagnosed with an eyelid SCC was 1.9 times greater in men >50 years of age (95% CI = 1.5–2.3%) [85].

### 3.4. Survival

Two studies examined the hazard ratio (HR) for sex and disease-free survival rates (DFS). The DFS male HR was 0.8 (95% CI: 0.3–2.5) *p* = 0.697; 1.01 (95% CI: 0.054–1.89) *p* = 0.972 for Kampel et al. [86] and Mooney et al. [87], respectively. Additionally, Tseng et al. found no significant difference in the DFS of lip cSCC between males and females 0.66 (95% CI: 0.22–1.94) *p* = 0.45. Two papers looked at 5-year overall survival (OS). Kampel et al. reported male HR for OS, 0.59 (95% CI: 0.3–1.2), *p* = 0.143 [26] and Kyrgidis et al. reported men vs. women, 71% vs. 66%, *p* = 0.106 [88].

Three studies examined sex differences in the five-year relative survival rate (RSR) in eyelid cSCC. Two of these studies found the five-year relative survival rate RSR to be higher in males than females. Karljalainen et al. found that the five-year RSR for males increased from 93.5% in 1967–1973 to 97.2% in 1974–1981 [89]. The RSR for females was 91.0% in 1967–1973 to 2.7% in 1974–1981. Jung et al. found that the RSR of males increased from 90.9% in 1993–1995 to 102.3% in 2011–2016 and increased from 84.4% in 1993–1995 to 99.0% in 2011–2016 in females [90]. Conversely, Hollstein et al. found no change in the five-year RSR across the study period; however, females had an RSR of 102%, whereas, for males, it was 94% [91].

The five-year RSR for males and females with ear cSCC was explored in four studies, two of which found males had higher RSR than females. The earliest data demonstrated that RSR increased for both males and females during the study period; however, it was consistently higher for males than their female counterparts (1967–1973: 83.1% and 54.2% in males and females, respectively; 1974–1981: 85.6% and 73.2% in males and females, respectively) [89]. Likewise, Iverson et al. found the RSR to increase over the reporting period; however, it remained higher in males than females (97.8% and 84.8%, respectively) [92]. Conversely, Robsahm et al. found the RSR was higher in males during their first reporting period of 1963–1969, with a rate of 0.87 in males and 0.33 in females; however, the reverse finding was found during their 2000–2011 reporting period where females were found to have better RSR than males (0.92 and 0.83, respectively) [80]. Hollstein et al. found the RSR consistent across males and females at 93% [91].

Two studies explored sex differences in the five-year RSR of cSCC of the scalp and both studies found that females had higher rates in females than males. Karjalaninen et al. found the RSR in males to increase from 70.8% in 1967–1973 to 80.2% in 1974 [89]. The RSR in females was 104.4% in 1967–1973 and decreased to 98.2% in 1974–1981. Hollstein et al. also found the RSR slightly higher in females than males, with 90% in females, and 89% in males [91].

## 4. Discussion

This systematic review of the past 30 years confirms that sexual disparities for the incidence of HNcSCC exist globally, with the current paper demonstrating that males made up 71% of all HNcSCC across the included studies. Since HNcSCC is the most common cancer, has metastatic potential and presents a great financial burden, we believe this disease requires greater public health priority. Metastatic cSCC holds a poor prognosis due to nodal/distant metastasis [5]. The national incidence of cSCC for men is significantly higher than that of women in Australia [2,4,77], although this figure is considered underreported [2]. A large longitudinal study in the U.S. demonstrated invasive SCC was more prevalent in men, including HNcSCC, after an 18-year follow-up [84]. Although cSCC of the eyelid had less of a strong predominance in males (56%), males accounted for 92% of all ear cSCCs. The data also shows that the estimated incidence rates of these locations are markedly different [10]. The cause for these differences in cancer location has been discussed in the literature.

It has long been thought that men are at a higher risk of HNcSCC due to occupational and lifestyle reasons, including the likelihood of performing outdoor work [92], having male-pattern baldness (scalp cSCC) [93], shorter haircuts (ear cSCC) [94] and not wearing lipsticks (external lip cSCC) [95]. Despite this, epidemiological studies adjusting for these risk factors still demonstrate that males suffer disproportionately for the prevalence of cSCCs and mortality across cancer subtypes [6,93]. For example, Duran et al. found poorer outcomes of non-scalp cSCC in men which potentially confounds the hypothesis that male-pattern baldness is a significant risk factor for poor outcomes in cSCC [94]. To date, increasing evidence suggest sex-specific differences in genetic and epigenetic factors contributing to disparity between the sexes for cancers in general [96] However, the details of these factors remain elusive for all cancers.

Interestingly, there were no studies which reported sex-compared genomic analysis that were included in this review. Genetic damage by ultraviolet B (UVB) radiation is considered an early event in cSCC carcinogenesis, and sex-specific differences have been observed in the mutational response to UVB damage. However, these differences have been observed mainly in animal studies. Male mice subjected to the same UVB levels developed more extensive and intrusive cutaneous tumours earlier than the female comparison group [6,95]. Notably, few studies have also observed differences in genomic profiles between men and women that may account for more aggressive carcinogenesis or a poorer response to cancer treatment in men. Li et al. identified large differences in mutation density and frequency of specific genes (BAP1, β-catenin) between the somatic mutation profiles of various tumours of men and women. They inferred that these might account for sex biases in DNA mismatch repair genes or microsatellite stability. Remarkably, sex biases were seen in genes controlling lipoprotein and sterol activity. A female-biased loss of genetic alterations in a specific gene (NCKAP5) was found in the head and neck (mucosal) SCC [14]. Significant sex differences in gene regulation controlling drug metabolism, possibly accounting for increased response to adjuvant chemotherapy and subsequent survival in colon adenocarcinoma [96]. Additionally, Dunford et al. have proposed the biallelic expression of six specific X-chromosome ‘escape from X-inactivation tumour suppressor’ [97] genes (e.g., ATRX, CNKSR2, DDX3X, KDM5C, KDM6A, and MAGEC3) in females as a protective factor against X-inactivation loss of function mutations [97]. One study also found promoter methylation (and subsequent repression of gene expression) of microRNA-137 (miR-137), which plays a role in the cell cycle, to be increased in females compared to males based on methylation-specific PCR analysis, although this was only limited to oral HNSCC [98].

Immunodeficiency and immunosuppression are risk factors for more aggressive cSCC, and that sexual dimorphism in immune system activation exists [15]. Assuming immunocompetence, females tend to have higher CD4+ T-cell counts and develop larger immune responses to infections and certain vaccines. Conversely, females are more susceptible to autoimmune diseases caused by immune overactivation [15,99]. Only one study has provided data detailing a sex-biased immune response in cSCC in mice, and no studies were found to prove a sexual bias in HNcSCC in humans. Budden et al. propose that the mutational and transcriptomic landscape of cancer differs by sex in that woman, assuming immunocompetence, may develop early strong immune activation and subsequently less aggressive primary cSCC [11]. The study involved exposing male and female mice to equal doses of 7,12-Dimethylbenz[a]anthracene (DMBA) to induce epidermal Hras mutations and topical tetradecanoyl-phorbol acetate (TPA) to stimulate inflammation and epidermal proliferation; followed by molecular and histological analyses of treated skin. Whilst DNA damage accumulated equally in the skin in both sexes, female skin demonstrated a stronger antitumour response following carcinogen exposure. Molecular analyses revealed higher levels of the cytokine IFN gamma (Ifng), a strong driver of antitumour activity in female skin. Importantly, a stronger overall transcriptomic response, including an increased upregulation of Cdkn2a (senescence-inducing tumour suppressor gene), was also observed in female skin. Histological analyses also revealed a larger infiltrate of CD4/CD8 T cells and CD1 antigen-presenting cells in female skin and—in contrast—more macrophages in male skin. Analyses of single-cell RNA from keratinocytes of cSCC-adjacent human skin supported these findings. They suggested a multifaceted immune response to UV damage involving T cells, IFNG and other cytokines, and antigen-presenting cells in female skin. Interestingly, this stronger immune response was reversed in female mice treated with prednisolone (induced immunosuppression), further supporting the immune system’s role in preventing cancer progression [6]. Castro et al. support this theory by being the first to present data showing higher CD4+ T cell counts in females and subsequently poorer MHC II-based driver mutation presentation, suggesting stronger immunoediting early on in tumourigenesis in younger and immunocompetent females [15].

Evidence for a role for the immune regulated resistance to cancer development and progression is most prominent in the study of cSCC in immune suppressed organ transplant recipients [100]. Cancer rates in organ transplant recipients are generally higher, and this rate is between 60–100 times greater than the age-matched population for cSCC. The extremely high tumour mutational burden seen in whole genomic analysis of metastatic cSCC [101] underpins the predilection for disease progression in those patients with impaired immune surveillance.

Concerning the interplay between sex and response to cancer treatment with immunotherapy, there is growing evidence detailing sex differences in response to immunotherapy treatment, where females tend to have poorer response rates than males. However, findings appear inconsistent depending on patient demographics and tumour type [37,39]. The recent report of findings from the MOUSEION-01 study comparing the effectiveness of checkpoint inhibitors across multiple cancer types showed a significant and reduced overall effect of therapy in females [102]. CSCC was not included in the series of trials reviewed. Still, anecdotal reports from practitioners involved in the use of immunotherapy in advanced or unresectable cSCC suggest a sex difference may exist, which warrants further examination. Castro et al. proposed that stronger immunoediting in early tumorigenesis in females leads to “immunologically invisible” antigens and subsequent poorer T-cell response against cancer cells as an explanation for poorer response to immune checkpoint blockade for the treatment of multiple cancers in females compared to males [15]. Conversely, sex differences have been observed in miRNA (X-linked small non-coding RNAs affected in tumorigenesis), where females express higher levels of miR-424 (322) which may reduce resistance and improve response to immunotherapy by inhibiting the PD-L1 immune checkpoint [96].

Limitations of this study include a racial bias, wherein most of the studies included were conducted on mostly Caucasian populations. HNcSCC is substantially a disease of fair-skinned people living in lower latitudes where UV exposure is greatest. A study in South Africa observed a population comprising mostly black individuals and covered a broad range of phototypes [103]. Still, it found that the incidence of cSCC for males was more than double that of females (20.8 vs. 8.5 per 100,000) [103]. This bias may also be partially accounted for by the expected higher prevalence of cutaneous malignancies in Caucasian populations. Nevertheless, more racial diversity should be included in future studies. The scope of this study is also acknowledged to be limited to HNcSCC, limiting its applicability to sex biases for other cancers.

Future research should focus on integrative multiomic analyses to improve our understanding of the molecular mechanisms driving sexual bias in carcinogenesis and progression in HNcSCC. Lopes-Ramos et al. proposes using network-based approaches where the interactions between altered genes and cellular components, rather than individual genes, are studied to provide better insight into mechanisms driving sex differences in cancer [96]. The effect of acquired immunodeficiency and whether it overcomes a female advantage in reduced incidence of HNcSCC should be undertaken. Analysis of different skin types and races to identify the beneficial impact of darker skin between sexes might also provide further clues to sex differences.

## 5. Conclusions

The evidence in the literature reviewed in this paper, whilst limited, underscores the sexual disparity in incidence and risk of metastatic progression of HNcSCC. Long-held beliefs of predominantly lifestyle factors driving a greater burden of cSCC in males appear over-simplified, with more recent evidence supporting a sex-linked immune mediated mechanism for reducing disease progression in females. The immune mediated pathway for reducing the incidence of cSCC is supported by the established rate—sixty times—of cSCC development in the immunosuppressed organ transplant recipient. Additionally, with a tumour mutational burden greater than any other cancer, mechanisms for DNA repair are likely to play a substantial role in disease incidence and progression. Improving our understanding of sex-specific protective and detrimental mechanisms in HNcSCC will better inform our preventative, therapeutic and prognosticative practices, and eventually improve clinical outcomes for all individuals—male and female.

## Figures and Tables

**Figure 1 cancers-14-05830-f001:**
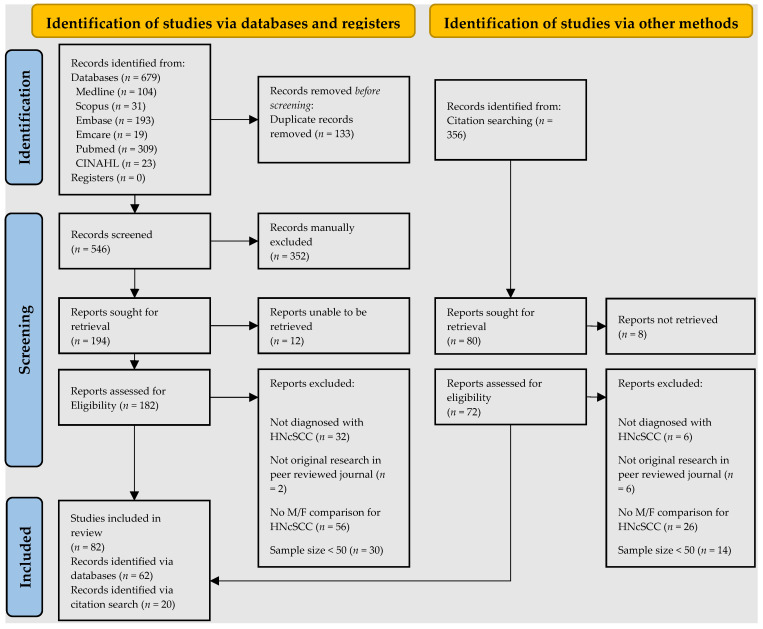
PRISMA flow diagram of the literature search.

**Figure 2 cancers-14-05830-f002:**
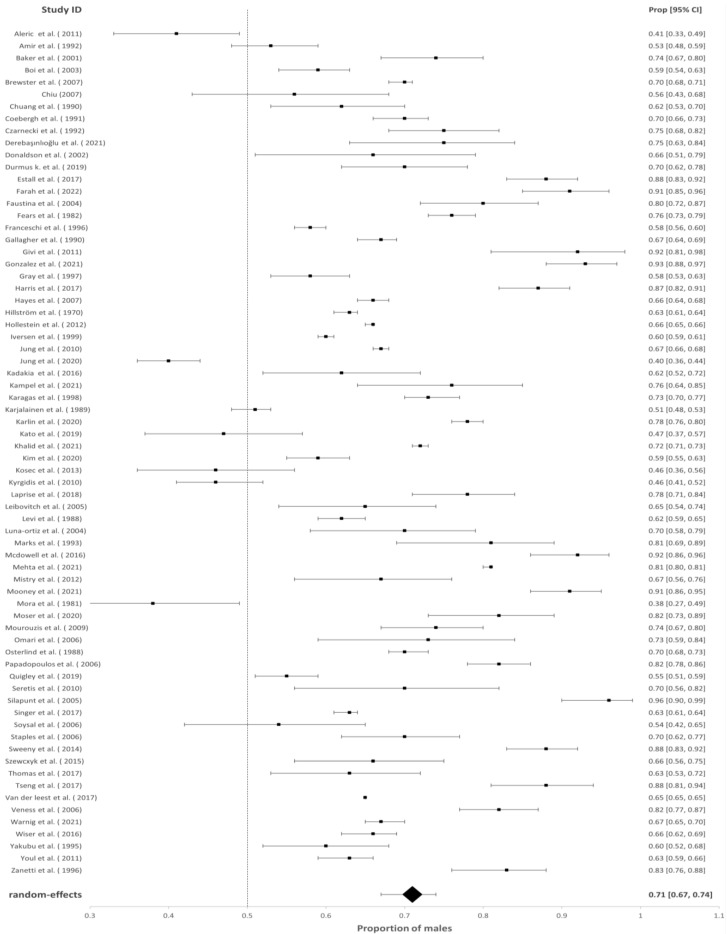
Proportion of males with cSCC of the total head and neck region [17,18,19,20,21,22,23,24,25,26,27,28,29,30,31,32,33,34,35,36,37,38,39,40,41,42,43,44,45,46,47,48,49,50,51,52,53,54,55,56,57,58,59,60,61,62,63,64,65,66,67,68,69,70,71,72,73,74,75,76].

**Figure 3 cancers-14-05830-f003:**
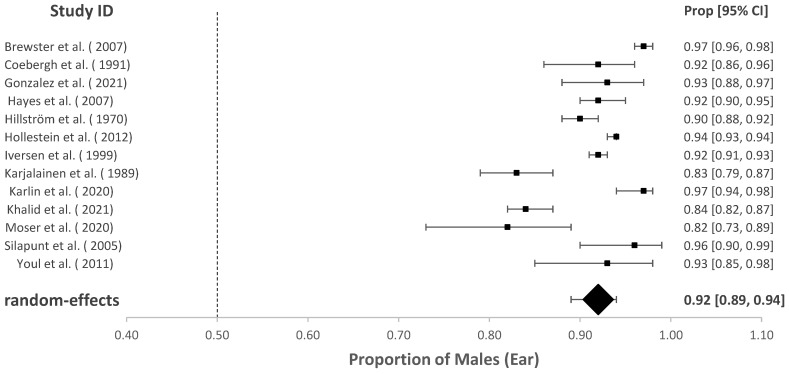
Proportion of males with cSCC of the ear [21,23,29,31,32,35,38,39,43,45,58,65,75].

**Figure 4 cancers-14-05830-f004:**
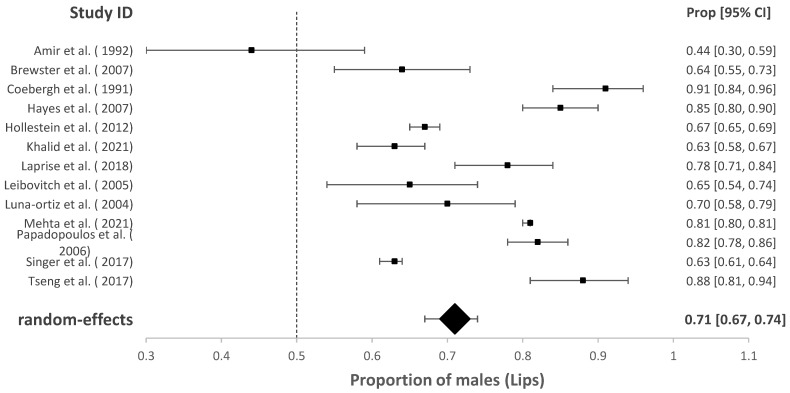
Proportion of males with cSCC of lips [17,21,23,35,38,45,48,49,51,54,62,70].

**Figure 5 cancers-14-05830-f005:**
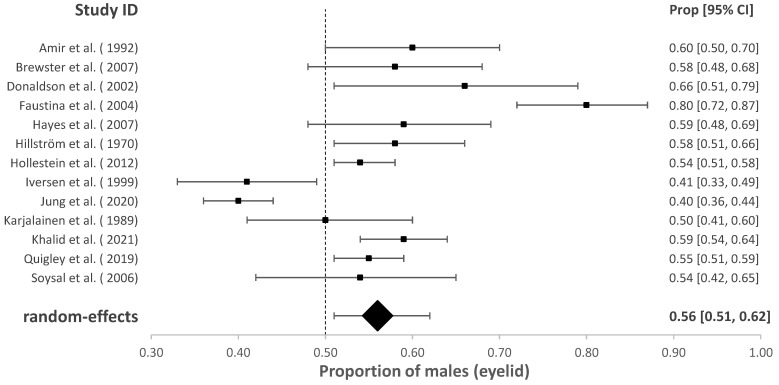
Proportion of males with cSCC of the eyelid [17,21,26,29,30,31,32,38,39,45,63,66].

## Data Availability

The data presented in this study are available on request from the corresponding author.

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
