# Peer review of "Sex Disparity for Patients with Cutaneous Squamous Cell Carcinoma of the Head and Neck: A Systematic Review"

_cancers, 2022, doi:10.3390/cancers14235830_

Round 1

Reviewer 1 Report

Thank you for the opportunity to review the review with the title “Sex disparity for patients with cutaneous squamous cell carcinoma of the head and neck: a systematic review”.

Even if sex disparity is an important topic overall, the article don´t point out the consequences of sex disparity in the particular for HNcSCC.

Author Response

Reviewer #1

  • Even if sex disparity is an important topic overall, the article don’t point out the consequences of sex disparity in particular for HNcSCC.

Our response: Thank you for highlighting this, we have added content to the discussion and to the conclusion to discuss the consequences of this sex disparity. We hope these changes satisfy this concern.

Reviewer 2 Report

tables will be demonstrative and they are defecient 

the paper need to more easy to be red

Author Response

Reviewer #2

2.1  Tables are deficient from the paper and they will be demonstrative/informative to the paper

Our response: We originally included two tables (summary of included articles and Risk of Bias) as supplementary materials. We felt that the size of these would detract from the flow of the manuscript and originally didn’t wish to include them in the main body of the paper, however, if the reviewers believe that one might be more suitable in the main text, we can include it there instead.

2.2  Please check through the paper for clarity and ease of readability.

Our response: Thank you for the suggestion, we have done a thorough grammar and spelling check to improve the flow of the paper and to fix issues with readability.

Reviewer 3 Report

This paper is about sex disparities in patients with HNcSCC. I think that this is a good topic often underestimated even by clinicians so this article could fill a knoledge gap. 

The manuscript is well written and the systematic review is well defined with an exhaustive litterature research as described in MM section. 

Discussion section is well written. I perceive a lack of deeply medical sound, but there is probably a lack of data in this regard. I probably prefer a deepening in correlation between immodeficiency, sex, therapies and patients response.  

I think that this is a good work and

Author Response

Reviewer #3

This paper is about sex disparities in patients with HNcSCC. I think that this is a good topic often underestimated even by clinicians so this article could fill a knowledge gap. 

The manuscript is well written, and the systematic review is well defined with an exhaustive literature research as described in MM section. I perceive a lack of deeply medical sound, but there is probably a lack of data in this regard.

Discussion section is well written. I think that this is a good work.

3.1  I probably prefer a deepening in correlation between immunodeficiency, sex, therapies and patients response.

Our response: Thank you for your comments and for the suggestion on how to improve the discussion. We have added more content to address your concerns and discuss the relationship of immunodeficiency, therapies and patient response with respect to gender differences.

Reviewer 4 Report

In beginning, I would like to emphasize that I have found the article very interesting and worth to be published in the Cancers. The manuscript contains very interesting and promising results. All parts of the manuscript are logically, clearly, and concisely described.

I recommend elaborating on the “conclusion” section. Perhaps an attempt to explain the existing differences would enrich the article.

Author Response

Reviewer #4

In beginning, I would like to emphasize that I have found the article very interesting and worth to be published in the Cancers. The manuscript contains very interesting and promising results. All parts of the manuscript are logically, clearly, and concisely described.

4.1  I recommend elaborating on the “conclusion” section. Perhaps an attempt to explain the existing differences would enrich the article.

Our response: Thank you for you comments, we have expanded on both the discussion and conclusion sections, tied in the content added in the discussion and made an effort to explain these differences.